# Number of transurethral procedures after non-muscle-invasive bladder cancer and survival in causes other than bladder cancer

**Lars Holmberg**[1,2]*, Oskar Hagberg[3], Christel Häggström[1,4], Truls Gårdmark[5], Viveka Ströck[6], Firas Aljabery[7], Staffan Jahnson[7], Abolfazl Hosseini[8], Tomas Jerlström[9], Amir Sherif[10], Karin Söderkvist[11], Anders Ullén[8,12], Mats Enlund[1,13], Fredrik Liedberg[3,14], Per-Uno Malmström[1]

1 Department of surgical Sciences, Uppsala, University, Uppsala, Sweden, 2 School of Cancer and Pharmaceutical Sciences, King's College London, London, United Kingdom, 3 Department of Translational Medicine, Lund University, Malmö, Sweden, 4 Department of Public Health and Clinical Medicine, Northern Register Centre, Umeå University, Umeå, Sweden, 5 Department of Clinical Sciences, Danderyd Hospital, Karolinska Institute, Stockholm, Sweden, 6 Department of Urology, Sahlgrenska University Hospital and Institute of Clinical Sciences, Sahlgrenska Academy, University of Gothenburg, Gothenburg, Sweden, 7 Division of Urology, Department of Clinical and Experimental Medicine, Linköping University, Linköping, Sweden, 8 Department of Pelvic Cancer, Theme Cancer, Karolinska University Hospital, Stockholm, Sweden, 9 Department of Urology, School of Medical Sciences, Faculty of Medicine and Health, Örebro University, Örebro, Sweden, 10 Department of Surgical and Perioperative Sciences, Urology and Andrology, Umeå University, Umeå, Sweden, 11 Department of Radiation Sciences, Umeå University, Umeå, Sweden, 12 Department of Oncology-Pathology, Karolinska Institute, Stockholm, Sweden, 13 Region Västmanland, Center for Clinical Research, Hospital of Västmanland, Uppsala University, Västerås, Sweden, 14 Department of Urology, Skåne University Hospital, Malmö, Sweden

* lars.holmberg@kcl.ac.uk

## Abstract

### Background

Previous research has associated repeated transurethral procedures after a diagnosis of non-muscle invasive bladder cancer (NMIBC) with increased risk of death of causes other than bladder cancer.

### Aim

We investigated the overall and disease-specific risk of death in patients with NMIBC compared to a background population sample.

### Methods

We utilized the database BladderBaSe 2.0 containing tumor-specific, health-related and socio-demographic information for 38,547 patients with NMIBC not primarily treated with radical cystectomy and 192,733 individuals in a comparison cohort, matched on age, gender, and county of residence. The cohorts were compared using Kaplan-Meier curves and Hazard ratios (HR) from a Cox regression models. In the NMIBC cohort, we analyzed the association between number of transurethral procedures and death conditioned on surviving two or five years.

**Data Availability Statement:** Data used in the present study was extracted from the research database BladderBaSe, which is based on the

Swedish National Registry of Urinary Bladder Cancer (SNRUBC) and linkage to several national health-data registers. The data cannot be shared publicly because the individual-level data contain potentially identifying and sensitive patient information and cannot be published due to legislation and ethical review restrictions (https:// etikprovningsmyndigheten.se). Use of the data from national health-data registers is further restricted by the Swedish Board of Health and Welfare (https://www.socialstyrelsen.se/en/) and Statistics Sweden (https://www.scb.se/en/) which are Government Agencies providing access to the linked healthcare registers. The data in in BladderBaSe is partly available in annual reports from the Swedish National Registry of Urinary Bladder Cancer (SNRUBC) and are also accessible online at https://statistik.incanet.se/ urinblasecancer/. Researchers can propose and apply for data and studies in BladderBaSe or SNRUBC using a standardized form. After approved application, the project data administrators can upload study-specific files with selected variables to a server for statistical analysis through remote access.

**Funding:** This work was supported by the Swedish Cancer Society (grant numbers CAN 2019/62 (LH) and CAN 2020/0709 (FL)), Swedish Research Council (2021-00859)(FL), Lund Medical Faculty (ALF)(FL), and Skåne County Council's Research and Development Foundation (FL). The funding sources had no role in the study design, data analyses, interpretation of the results, or writing of the manuscript. Swedish Cancer Society: Cancerfonden.se Swedish Research Council: vr.se Lund Medical Faculty: lu.se Skåne County: Skane. se.

**Competing interests:** The authors have declared that no competing interests exist.

## Results

Overall survival and survival from causes other than bladder cancer estimated with Kaplan-Meier curves was 9.3% (95% confidence interval (CI) (8.6%-10.0%)) and 1.4% (95% CI 0.7%-2.1%) lower respectively for the NMIBC cohort compared to the comparison cohort at ten years. In a Cox model adjusted for prognostic group, educational level and comorbidity, the HR was 1.03 (95% CI 1.01–1.05) for death from causes other than bladder cancer comparing the NMIBC cohort to the comparison cohort. Among the NMIBC patients, there was no discernible association between number of transurethral procedures and deaths of causes other than bladder cancer after adjustment. The number of procedures were, however, associated with risk of dying from bladder cancer HR 3.56 (95% CI 3.43–3.68) for four or more resections versus one within two years of follow-up.

## Conclusion

The results indicate that repeated diagnostic or therapeutic transurethral procedures under follow-up do not increase of risk dying from causes other than bladder cancer. The modestly raised risk for NMIBC patients dying from causes other than bladder cancer is likely explained by residual confounding.

## Introduction

Recent findings question if routines for follow-up of individuals with non-muscle invasive bladder cancer (NMIBC) should be revised. Two reports hypothesize that repeated diagnostic or therapeutic transurethral procedures for local recurrences in individuals with NMIBC are associated with increased risk of death of causes other than bladder cancer. They found an inferior relative survival in individuals with NMIBC compared to a background population [1] and a higher risk of death associated with the number of transurethral resections [2]. If repeated interventions requiring anaesthesia in a group with many frail individuals iatrogenically causes deaths, this would necessitate not only a review of follow-up routines for NMIBC but also of other follow-up programs that include repeated anaesthesia. However, the cited papers reported overall survival and did not account for the contribution of bladder cancer death, which is necessary to understand the generalizability of the results; the reported five-year risk of progression in subgroups of NMIBC patients range from almost zero to 40% [3]. Furthermore, these reports did not account for that the number of follow-up procedures that an individual is exposed to is associated with severity of the bladder cancer and is an exposure that takes place when the follow-up underlying survival analyses already has started.

We tested the hypothesis forwarded in the cited papers by investigating the overall and disease-specific risk of death in a Swedish population-based cohort of patients with NMIBC. We utilized in the Bladder Cancer Data Base Sweden (BladderBaSe) 2.0 [4] with an individually matched comparison cohort and adjusted for comorbidity and socioeconomic factors and accounted for that the number of resections is a time-dependent variable.

## Material and methods

### Data source

BladderBaSe 2.0 is based on individuals with bladder cancer and cancer of the upper urinary tract consecutively reported to the Swedish National Register for Urinary Bladder Cancer

(SNRUBC) [5]. SNRUBC is a clinical database for audit and research. All treatment facilities for urological cancer in Sweden report to the register. The entries in SNRUBC are compared to the Swedish Cancer Register (SCR) for validation and check of coverage. Reporting to the SCR is mandated by law and information is actively sought for patients in the SCR missing in the SNRUBC. For 2017–2019 the capture for bladder cancer in SNRUBC compared to SCR was >98%. The SNRUBC holds extensive data on tumour characteristics and given treatment.

BladderBaSe 2.0 is a research data base and a further expansion of BladderBaSe 1.0 [4]. Using the Swedish National Personal Identification Number, the database links individuals registered in SNRUBC between 1 Jan 1997 and 31 Dec 2019 to a number of national registers. For the purpose of this study we used the linkage to: The Swedish Household Census and the Register of Total Population and Population Changes for vital status and data on immigration and emigration; the Longitudinal Integration Database for Health Insurance and Labour Market Studies (LISA) for information on socioeconomic status; the SCR for cancer occurrences prior to or after diagnosis of bladder cancer; the Patient Register for out- and inpatient interventions and diagnoses as exposures and for assigning Charlson Co-morbidity Index (CCI) [6]; and the Cause of Death Register.

In addition to the previous version, BladderBaSe 2.0 also contains a matched comparison cohort sampled from the background population with individuals free from cancer in the urinary tract at the time of diagnosis of the index case. To build the comparison cohort, five controls for each index case were randomly selected from the background population without replacement at date of diagnosis of the index case. Controls were matched for sex, year of birth and county of residence and were allowed to later develop cancer in the urinary tract, hence one individual could first be a member of the comparison cohort and later a case, with corresponding new controls allocated. The comparison cohort was linked to the same data sources as the index cases.

## Participants

We applied the following inclusion criteria to identify the participants in this study: Patients had to be diagnosed with a NMIBC stage Ta, Cancer in situ (CIS) or T1, having information on grade and not undergoing radical cystectomy as primary treatment (the NMIBC cohort). There were no losses to follow-up. The corresponding comparison cohort was included with the individual matching intact.

## Exposure and co-variates

The exposure of interest is follow-up after NMIBC not treated with radical cystectomy as primary treatment, and within that follow-up the number of in- or outpatient urological transurethral diagnostic or therapeutic procedures after first diagnosis for individuals with NMIBC. We identified these procedures by entries in the Patient Register of the Swedish Classification of Healthcare Procedures (KVÅ) codes for transurethral resection of bladder tumor or biopsy (KCD02/KCD96 and UKC05/KCB00 respectively), transurethral surgery of the prostate and bladder neck (KED98/KED22/KCH42), and transurethral evacuation of blood clots (KCW98/KCC22). We further stratified for prognostic group at diagnosis using stage (Ta, CIS, T1) and further using grade to separate Ta tumors (TaG1 and TaG2/G3) into subclasses in accordance with the Swedish National Guidelines for adjuvant instillation therapy and more frequent follow-up in high-risk groups. Histopathologic classification of tumor grade was performed according to WHO 73 grading system from 1997 to 2002 and according to WHO 99 from 2003 and onwards [7]. We used educational level as an indicator of socio-economic status.

Educational level was categorized as low (≤9 years of school), intermediate (10–12 years) and high (≥13 years), corresponding to mandatory school, high school and college or university.

We used the Charlson Comorbidity Index (CCI) [6] to assess co-morbidity. CCI was calculated based on diagnoses reported to the Patient Register from 10 years before diagnosis or to the index date of the case for the corresponding members of the comparison cohort. The Patient Register includes data on all procedures, (KVÅ codes), and all diagnoses, coded according to the International Classification of Diseases (ICD) 9th edition or ICD-10, for all patients in Sweden.

We used the adaptation of CCI to Swedish registries made by Ludvigsson et al. [8] based on ICD9 and ICD10 codes. In Ludvigsson's adaptation, CCI is a sum of points given for the following conditions, with points in parenthesis: Cerebrovascular disease (1), Congestive heart failure (1), Chronic obstructive pulmonary disease (1), Dementia (1), Diabetes (1), Mild liver disease (1), Myocardial infarction (1), Other chronic pulmonary diseases (1), Peripheral vascular (1), Rheumatic disease(1), Ulcer (1), Hemiplegia (2), Severe kidney disease (2), Severe liver disease (3), AIDS (6), Malignancy (2), Metastatic cancer (6).

## Outcomes

Outcomes were death of any cause, death of causes other than bladder cancer as underlying or contributing cause of death, and death of bladder cancer as underlying or contributing cause of death. Causes and dates of death were retrieved from the Causes of Death Register. Individuals were censored at emigration, at the event of cystectomy or radiotherapy with curative intent, or if diagnosed with a cancer of the upper urinary tract.

The study was approved by the Research Ethics Board of Uppsala University, Sweden (Ref no. 2015/277) and Swedish Ethical Review Authority (Ref no. 2019–03574 and 2020–05123) that waived the informed consent requirement.

## Statistical methods

In the comparison between the NMIBC cohort and their matched comparison cohort we used survival analyses with Kaplan-Meier curves with a log-rank test to assess differences between the curves. Date of bladder cancer diagnosis for individuals in the NMIBC cohort and the corresponding index date for individuals in the comparison cohort were used as start of follow up. The Kaplan-Meier analyses were stratified for prognostic group (TaG1, TaG2/G3, CIS, T1). Furthermore, we used stratified Cox Proportional Hazards models as a regression tool, where each case formed a stratum together with its corresponding members of the comparison cohort, matched for sex, year of birth and county of residence. We adjusted the models for prognostic group, CCI and educational level but not for the matching factors. The models were repeated for all outcomes.

To attain high statistical precision, the study used all the available information in the register during the study period. With, for example, 2000 events, a hazard ratio of 1.2 is detectable under standard assumptions (80% power, two-sided 5% significance level).

We analyzed the association between number of transurethral procedures only patients in the NMIBC cohort, not using the comparison cohort. These procedures are performed after diagnosis and thus during the follow-up creating the person-years in the denominator. To account for this, we used the first 2 and then the first 5 years of follow-up to count the number of procedures and used only the time after this period as follow-up time in the survival analysis. These analyses were performed for all outcomes. These analyses were adjusted for prognostic group, CCI, educational level and age.

As a sensitivity analysis, we used a Cox model with time dependent covariates, where an individual entered new time intervals when our defined limits for the number of procedures were reached. In this way, all follow-up time can be used. However, this model assumes that the effect depends directly on the number of procedures, irrespectively of when a procedure occurred, which is an assumption that is likely to be violated during follow-up for a progressive disease.

For all statistical analyses the R statistical package was used [9].

## Results

### Study population and baseline characteristics

We identified 38,547 patients registered with TaG1, TaG2/G3, CIS or T1 and not undergoing primary radical cystectomy to be included in the NMIBC cohort. Table 1 shows the baseline characteristics of the patients in the NMIBC cohort and their 192,733 age-, sex- and county-matched individuals in the comparison cohort. Twenty-four percent of the study population were women and 61.6% were over age 70. There was only a small shift for the educational level

**Table 1. Baseline characteristics and the number of transurethral procedures during 2 and 5 years of follow-up for the cohort of non-muscle invasive bladder cancer (NMIBC) patients and their matched comparison cohort.**

|  |  | NMIBC-cohort N = 38547 (16.7%) | Comparison cohort N = 192733 (83.3%) |
|---|---|---|---|
| **Sex** | **M** | 29303 (76.0%) | 146513 (76.0%) |
|  | **F** | 9244 (24.0%) | 46220 (24.0%) |
| **Age** | **<50** | 1326 (3.4%) | 6616 (3.4%) |
|  | **50–69** | 13308 (34.5%) | 66591 (34.6%) |
|  | **70+** | 23913 (62.0%) | 119526 (62.0%) |
| **Education** | **Low** | 15797 (41.0%) | 77840 (40.4%) |
|  | **Middle** | 14489 (37.6%) | 70000 (36.3%) |
|  | **High** | 7384 (19.2%) | 40270 (20.9%) |
|  | **Missing** | 877 (2.3%) | 4623 (2.4%) |
| **Resections within 2 years among the subjects with more at least 2 years follow-up** | **0** | 0 (0.0%) | 168642 (99.1%) |
|  | **1** | 13270 (39.0%) | 1187 (0.7%) |
|  | **2–3** | 15491 (45.5%) | 261 (0.2%) |
|  | **4 or more** | 5263 (15.5%) | 28 (0.0%) |
| **Resections within 5 years among the subjects with at least 5 years follow-up** | **0** | 0 (0.0%) | 135107 (98.0%) |
|  | **1** | 8776 (31.8%) | 2084 (1.5%) |
|  | **2–3** | 11021 (40.0%) | 534 (0.4%) |
|  | **4 or more** | 7777 (28.2%) | 143 (0.1%) |
| **CCI** | **0** | 20673 (53.6%) | 122983 (63.8%) |
|  | **1** | 6272 (16.3%) | 26663 (13.8%) |
|  | **2** | 6606 (17.1%) | 24950 (12.9%) |
|  | **3** | 2681 (7.0%) | 9962 (5.2%) |
|  | **3+** | 2315 (6.0%) | 8170 (4.2%) |
|  | **Missing** | 0 (0.0%) | 5 (0.0%) |
| **T/G group** | **Comparison cohort** | - | 192733 (100.0%) |
|  | **TaG1** | 12642 (32.8%) | - |
|  | **TaG2/TaG3** | 12959 (33.6%) | - |
|  | **CIS** | 1118 (2.9%) | - |
|  | **T1** | 11828 (30.7%) | - |

to be lower in individuals with NMIBC compared to the comparison cohort. In the NMIBC cohort, 53.6% had a CCI of 0, while the corresponding proportion for the comparison cohort were 63.8%. Patients with TaG1, TaG2/TaG3 and T1 were three similarly large groups constituting 97% of all, and the 1118 patients with CIS making up 2.9%. Over 50% of patients with NMBIC underwent two or more transurethral procedures during two years after diagnosis. Of the procedures, 84% were coded as transurethral resections of locally recurrent disease (KCD02). Some individuals in the comparison cohort– 0.8% within two years after index date–underwent transurethral procedures for diagnoses other than bladder cancer.

## Overall survival

Overall survival was lower in the NMIBC cohort than in the comparison cohort throughout the follow-up period and differed 9.3% (95% confidence interval (CI) (8.6%-10.0%)) percentage points at ten years (Table 2). The HR for death of all causes in the NMIBC cohort versus the comparison cohort and based on the entire follow-up information was 1.43 (95% CI 1.40–1.46). An adjustment for prognostic group, educational level and CCI attenuated the HR to 1.35 (95% CI 1.32–1.37) (Table 2).

Fig 1 shows Kaplan-Meier curves for overall survival by prognostic groups. For all groups, overall survival was lower for the NMIBC cohort versus their matched comparison cohort. The difference was more pronounced for patients with T1 with an unadjusted HR of 1.86 (95% CI 1.81–1.92) as compared to a HR of the order of 1.2 to 1.3 comparing patients with TaG1 and TaG2/TaG3 to their respective comparison cohorts. The corresponding HR was 1.47 for patients with CIS, but with a broader confidence interval (95% CI 1.32–1.63).

## Mortality from causes of death other than bladder cancer

Survival from causes of death other than bladder cancer was lower in the NMIBC cohort than in the comparison cohort, the absolute difference at ten years was 1.4% (95% CI 0.7%-2.1%) percentage points. An unadjusted HR using the entire follow-up period was 1.10 (95% CI 1.08–1.12) and when adjusted for prognostic group, educational level and CCI HR was 1.03 (95% CI 1.01–1.05) (Table 2).

Fig 2 shows Kaplan-Meier curves of mortality from causes of death other than bladder cancer by prognostic groups. In all prognostic groups, the mortality was statistically significantly higher in the NMIBC cohort than for the corresponding comparison cohorts, however with

**Table 2. Kaplan-Meier estimates of overall survival and survival for causes other than bladder cancer for the cohort of non-muscle invasive bladder cancer (NMIBC) patients and their matched comparison cohort.**

| | | Overall survival | | Other causes | |
|---|---|---|---|---|---|
| | | NMIBC cohort | Comparison cohort | NMIBC cohort | Comparison cohort |
| **Survival (% after number of years)** | **2** | 86.8 | 91.4 | 91.1 | 91.4 |
| | **5** | 70.9 | 78.7 | 78.1 | 78.9 |
| | **10** | 50.0 | 59.3 | 58.1 | 59.5 |
| | **20** | 22.6 | 29.6 | 28.2 | 29.8 |
| **HR with95% CI** | **Unadjusted** | 1.43 (1.40–1.46) | 1 | 1.10 (1.08–1.12) | 1 |
| | **Adjusted for prognostic group, education and CCI** | 1.35 (1.32–1.37) | 1 | 1.03 (1.01–1.05) | 1 |

Hazard ratios (HR) compare the NMIBC cohort to the comparison cohort. Hazard ratios are derived from a Cox model stratified for the matching on sex, age, and county of residence. CCI = Charlson comorbidity index.

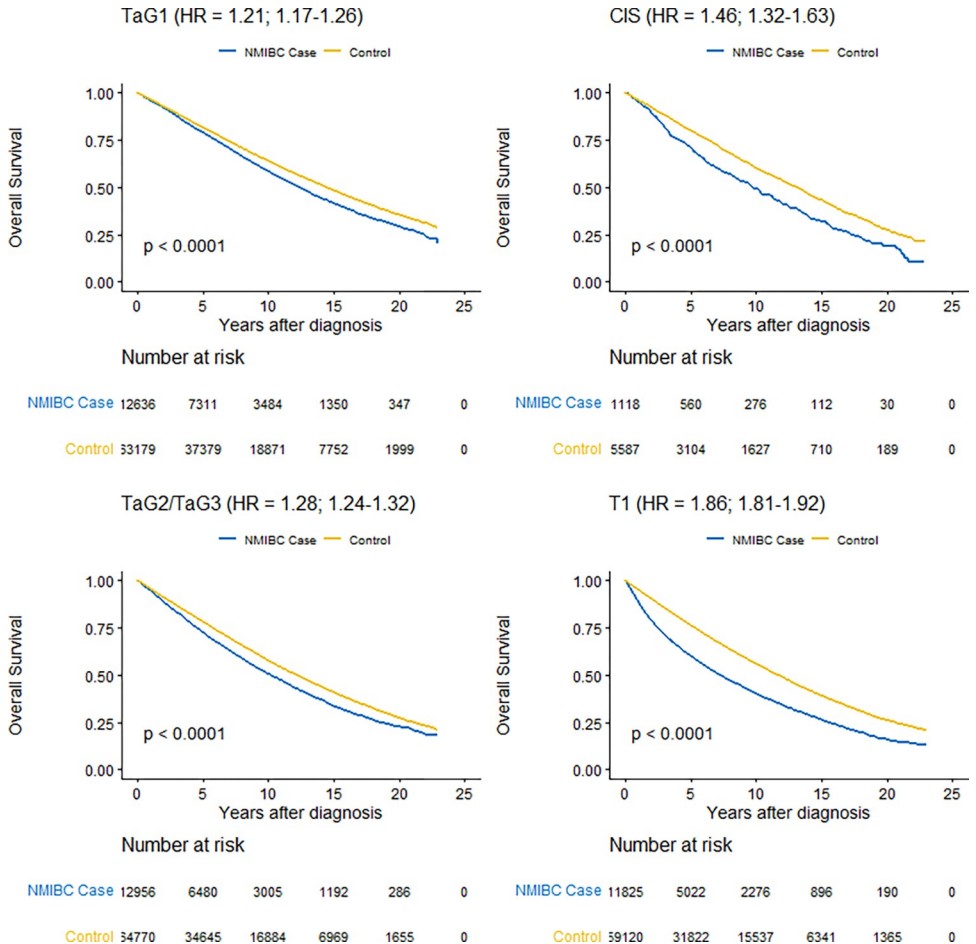

**Fig 1. Kaplan-Meier estimates of overall survival in the non-muscle invasive bladder cancer cohort (NMIBC) by prognostic group compared to their respective matched comparison cohorts with log-rank test of statistical significance and unadjusted hazard ratio (HR) with 95% confidence interval from a Cox proportional hazards model.**

lower unadjusted HRs than for overall mortality varying from 1.04 (95% CI 1.00–1.07) to 1.18 (95% CI 1.14–1.22) for patients with TaG2/TaG3 and patients with T1, respectively.

## Association between number of resections and mortality

To account for the time-dependence of number of resections, we analysed risk of death in the NMIBC cohort separately conditioned on having lived two years and with the number of resections within those two years as exposure. A univariate Cox proportional hazards model of overall mortality showed a trend for increasing risk with more resections with a HR of 1.41 (95% CI 1.34–1.48) for four or more resections versus one (Table 3). The same model for risk of death from causes other than bladder cancer yielded a HR of 1.08 (95% CI 1.02–1.14) for four and more resections versus one. Adjustment for prognostic group, CCI, educational level and age attenuated the results and the HR was close to unity for deaths from causes other than bladder cancer. There was an association between number of resections and bladder cancer mortality both in uni- and multivariate models with a more than 3-fold increased risk for four or more resections versus one even after adjustment (Table 3). The same analyses were repeated conditioned on having lived five years and the results showed the same pattern (Table 3).

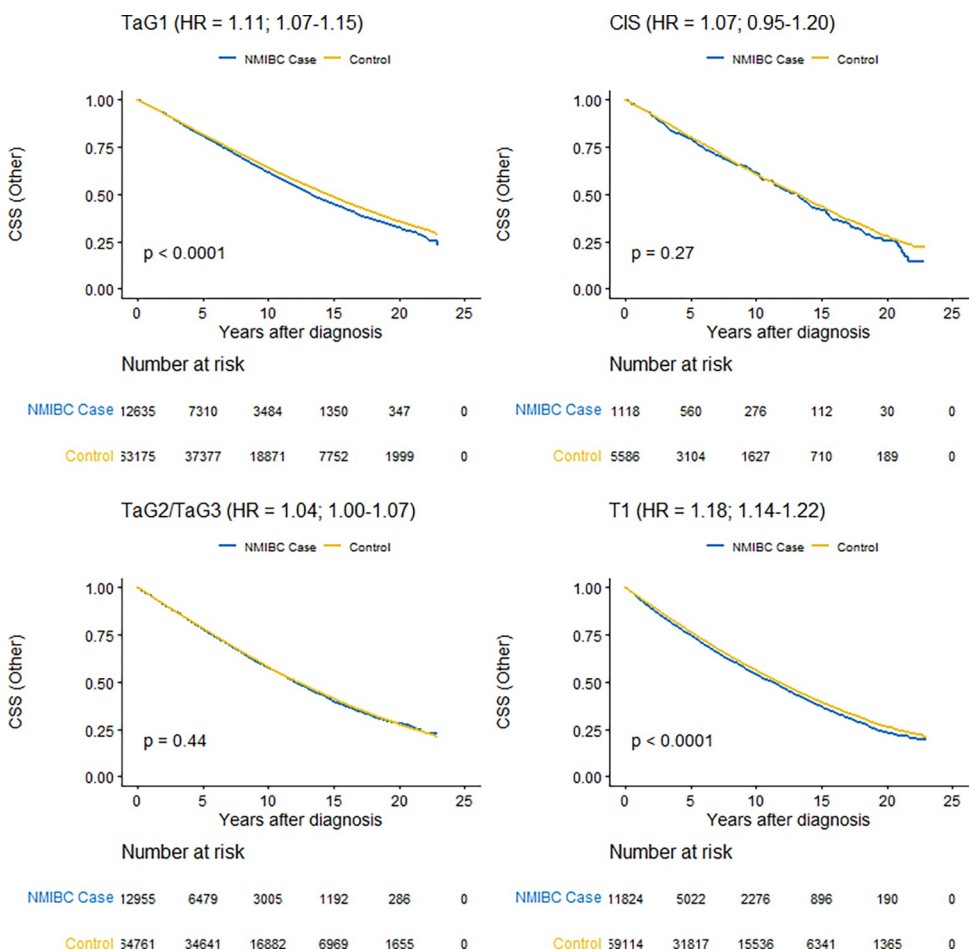

**Fig 2. Kaplan-Meier estimates of survival from causes other than bladder cancer in the non-muscle invasive bladder cancer cohort (NMIBC) by prognostic group compared to their respective matched comparison cohorts with log-rank test of statistical significance and unadjusted hazard ratio (HR) with 95% confidence interval from a Cox proportional hazards model.**

**Table 3. Hazard ratios (HR) with 95% confidence intervals from a Cox proportional hazards model estimating risk of dying from all causes, of causes other than bladder cancer and of bladder cancer associated with number of postoperative transurethral procedures in the NMIBC cohort.**

| | N | #Dead (%) | Overall survival | | Other causes | | Bladder cancer | |
|---|---|---|---|---|---|---|---|---|
| **Conditioned on 2 years survival** | | | Unadjusted | Adjusted | Unadjusted | Adjusted | Unadjusted | Adjusted |
| 1 | 10940 | 4467(40.8%) | 1 | 1 | 1 | 1 | 1 | 1 |
| 2–3 | 13623 | 5896(43.3%) | 1.16 (1.13–1.20) | 1.06 (1.01–1.10) | 1.06 (1.02–1.10) | 0.98 (0.94–1.03) | 2.25 (2.13–2.36) | 1.79 (1.67–1.91) |
| 4 or more | 4918 | 2517(51.2%) | 1.40 (1.35–1.45) | 1.28 (1.23–1.33) | 1.07 (1.01–1.12) | 1.01 (0.96–1.07) | 4.67 (4.55–4.79) | 3.56 (3.43–3.68) |
| **Conditioned on 5 years survival** | | | | Trend: P<0.001 | | Trend: P = 0.840 | | Trend: P<0.001 |
| 1 | 5632 | 2162(38.4%) | 1 | 1 | 1 | 1 | 1 | 1 |
| 2–3 | 7708 | 3075(39.9%) | 1.11 (1.06–1.17) | 1.05 (0.99–1.10) | 1.06 (1.00–1.11) | 1.00 (0.95–1.06) | 2.13 (1.92–2.35) | 1.90 (1.68–2.12) |
| 4 or more | 6022 | 2742(45.5%) | 1.29 (1.23–1.34) | 1.23 (1.17–1.29) | 1.06 (1.00–1.12) | 1.03 (0.97–1.09) | 5.19 (4.99–5.39) | 4.62 (4.41–4.82) |
| | | | | Trend: P<0.001 | | Trend: P = 0.340 | | Trend: P<0.001 |

#Dead shows the total number of deaths (all causes) in respective category. Analyses are conditioned on surviving two respectively five years, counting the number transurethral procedures within the respective interval. Adjusted models account for prognostic group, age, educational level and CCI.

In a sensitivity analysis we applied a model with the number of transurethral procedures as a time-dependent co-variate, adjusted for prognostic group, educational level, CCI and age. There was no discernible trend for an association between number of procedures and mortality from causes other than bladder cancer.

## Discussion

Overall survival was lower in NMIBC patients of all prognostic groups not undergoing primary cystectomy compared to a matched comparison cohort sampled from the background population. In a Cox model adjusted for prognostic group, educational level and comorbidity, the HR was 1.03 (95% CI 1.01–1.05) for death from causes other than bladder cancer comparing the NMIBC cohort to the comparison cohort. In the NMIBC cohort, there was no discernible association between number of resections and deaths of causes other than bladder cancer after adjustment for prognostic group, CCI, educational level, and age.

The lower overall survival among all prognostic groups of NMIBC patients was mainly due to a higher risk of dying from bladder cancer as cause of death, especially for patients with clinical T1 tumours. This finding demonstrates that bladder cancer mortality cannot be ignored when interpreting patterns of survival in NMIBC. Assumptions about the contribution of bladder cancer death to overall mortality in a cohort of NMIBC patients need to consider the time at risk and the stage-distribution within the NMIBC-group, as demonstrated when recently defining a new EAU "very high risk" prognostic NMIBC-group [3]. In this study, we were able to use individual data on cause-specific mortality.

There was still a higher cumulative mortality in the NMIBC cohort compared to the comparison cohort when only deaths other than from bladder cancer were considered. The size of the quantitative estimates of this higher risk is likely related to the high exposure of tobacco smoking among NMIBC patients. One out of two patients diagnosed with bladder cancer in Sweden 2019–2020 were former (33%) or current (17%) smokers [10]. In contrast, the level of current smokers was 7% among both women and men in the Swedish population the same years [11]. We have no individual data on smoking in this study, but the elimination of the higher risk when we adjusted for CCI and educational level–both associated with smoking [12]–further indicates that the crude estimates are confounded by smoking. Also, the associations between BMI, hypertension and bladder cancer risk seen both in smokers and never-smokers adds further to possible confounding in the comparison of mortality between NMIBC patients and the background population [13].

A corroboration of the hypothesis forwarded by Schmidt Erikson et al. that repeated transurethral procedures under general anaesthesia leads to higher mortality could have influenced current practice for follow-up in NMIBC patients. However, the worse survival in causes other than bladder cancer we found is most likely due to residual confounding and not the management per se. Furthermore, we could not find evidence for that number of procedures was associated with mortality from causes other than bladder cancer. As a contrast, we found a strong association between number of procedures and risk of death from bladder cancer, showing that analyses of the association between mortality and number of procedures during follow-up need to account for severity of the disease prompting these procedures.

Schmidt Erikson et al. [2] assume that the majority of the procedures used as exposure in their study were performed under general anaesthesia which could induce repeated physiologic stress. In our setting, we had no indicator variable for if the procedure was done under local, epidural, spinal or general anaesthesia. Thus, if general anaesthesia is the detrimental factor rather than the general stress and trauma from repeated procedures, such an effect may be partly hidden in our study. However, a higher risk would still likely be detected in patients

with many procedures, which we did not find. Within 30 days, epidural and spinal anaesthesia have less general and potentially life-threatening side-effects than general anaesthesia [14]. The interpretation of the evidence is not unequivocal and there is no conclusive evidence for a difference between the anaesthesiologic methods on mortality and morbidity in the longer run [14]. A review of neuraxial versus general anaesthesia specifically in urological surgery could not substantiate a difference in mortality within the first year of follow-up [15]. Any observational comparison between different anaesthesiologic methods over long-time is likely to be confounded by the indication for the method used, general anaesthesia being associated with worse conditions and more severe disease than neuraxial methods.

Another ongoing debate is if the anaesthesiologic method used may influence cancer progression. Different effects on recurrence-free survival after TURB performed under general and spinal anaesthesia have been described in low- and intermediate-risk NMIBC and high-risk NMIBC, where improved recurrence-free survival was reported in patients with high-risk NMIBC operated under spinal anaesthesia [16]. However, the problem is further complicated by that not only neuraxial methods may differ from general anaesthesia in effect, but different forms of general anaesthesia may also influence cancer progression differently. Current evidence is contradicting and results of RCTs are awaited [17, 18]. Inhalation anaesthesia, likely to have been most prevalent under the shorter procedures for NMIBC patients, are postulated to be more harmful than intravenous anaesthesia in terms of cancer progression. For a specific method of anaesthesia used in a subgroup to influence our results markedly, the association between that anaesthesia method and mortality would have to be very strong, a theory for which there is no current evidence in the literature.

## Strengths and limitations

This study is based on a national data set with high coverage and without losses to follow-up. The statistical precision is very high also for long-term results and even small differences between groups can be ascertained with narrow confidence intervals. We can account for comorbidity and educational level. We also accounted for that the exposure-variable number of procedures varies within the time-window *after* start of follow-up and must be treated as a time-dependent co-variate; we accounted for time-dependence with an analysis conditioned on having lived two or five years and used a model approach as a sensitivity analysis with very similar results. We did not use a model with time-dependent co-variates as our main model because such a model assumes that the exposure occurs at random and is unrelated to health status, and thus that there is a continuity over time of the effect of the number of resections. This assumption is unlikely to hold fully in our setting studying a progressive disease [19].

A major drawback for a study of causes of death other than bladder cancer is that we lack information on tobacco-smoking and other personal life-style factors relevant for bladder cancer risk and progression. Adjustment for CCI and educational status can partly compensate for this, but residual confounding is likely. We lack information about the use of office-based fulguration under local anaesthesia of small local recurrences performed at follow-up cystoscopies. However, there is no clear mechanism why any distribution of such procedures would have concealed an association between transurethral procedures and survival in causes other than bladder cancer.

The clinical staging of NMIBC entails misclassification. In this study, however, there were no signs of interaction by prognostic group. We could not stratify the analyses according to the EAU 2021 risk score for progression [3] due to that SNRUBC only recently included information on tumour size and number of tumours. However, we used subclasses that informs decisions for adjuvant instillation therapy and more frequent follow-up in accordance with the

Swedish National Guidelines. Furthermore, clinical stage is the basis for clinical decision making and therefore relevant as a point of departure in weighing benefits and risks in a follow-up program.

## Conclusion

These data indicate that repeated diagnostic or therapeutic transurethral procedures under anaesthesia do not increase mortality, which is important information for those settings where such a follow-up program can prevent bladder cancer progression and associated morbidity and mortality. However, this does not preclude the use of office-based fulguration under local anaesthesia and a constant evaluation of the risk/benefit of different follow-up procedures should continue in clinical practice. Around 20% of all Swedish NMIBC patients not undergoing primary cystectomy underwent 4 or more procedures, which makes up for a substantial number of individuals put under stress and for a substantial health-care resource utilization.

## Author Contributions

**Conceptualization:** Lars Holmberg, Christel Häggström, Fredrik Liedberg, Per-Uno Malmström.

**Data curation:** Oskar Hagberg, Christel Häggström, Truls Gårdmark, Viveka Ströck, Firas Aljabery, Staffan Jahnson, Abolfazl Hosseini, Tomas Jerlström, Amir Sherif, Karin Söderkvist, Anders Ullén.

**Formal analysis:** Lars Holmberg, Oskar Hagberg, Christel Häggström, Fredrik Liedberg, Per-Uno Malmström.

**Funding acquisition:** Lars Holmberg, Fredrik Liedberg.

**Investigation:** Viveka Ströck, Firas Aljabery, Staffan Jahnson, Abolfazl Hosseini, Tomas Jerlström, Amir Sherif, Karin Söderkvist, Anders Ullén.

**Methodology:** Lars Holmberg, Oskar Hagberg, Mats Enlund, Per-Uno Malmström.

**Project administration:** Lars Holmberg.

**Resources:** Lars Holmberg, Truls Gårdmark.

**Supervision:** Lars Holmberg, Mats Enlund, Fredrik Liedberg.

**Writing – original draft:** Lars Holmberg.

**Writing – review & editing:** Oskar Hagberg, Christel Häggström, Truls Gårdmark, Viveka Ströck, Firas Aljabery, Staffan Jahnson, Abolfazl Hosseini, Tomas Jerlström, Amir Sherif, Karin Söderkvist, Anders Ullén, Mats Enlund, Fredrik Liedberg, Per-Uno Malmström.

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
