## [Decision Letter · Decision Letter 0]

2 Aug 2022

PONE-D-22-17365Number of transurethral procedures after non-muscle-invasive bladder cancer and survival in causes other than bladder cancerPLOS ONE

Dear Dr. Holmberg,

Thank you for submitting your manuscript to PLOS ONE. After careful consideration, we feel that it has merit but does not fully meet PLOS ONE’s publication criteria as it currently stands. Therefore, we invite you to submit a revised version of the manuscript that addresses the points raised during the review process.

We look forward to receiving your revised manuscript.

Kind regards,

Kuo-Cherh Huang

Academic Editor

PLOS ONE

Journal Requirements:

Additional Editor Comments:

Dear Prof. Holmberg,

We appreciate your submission to PLoS ONE. In addition to the reviewers’ comments, I would like to bring up a couple of points required your clarifications:

1. Firstly, I feel confused as regards the comparison cohort was matched on what variables exactly, as there were inconsistent statements in your paper --

(a). P. 5, lines 103-104, “Controls were matched for sex, year of birth and county of residence”.

(b). P. 7, lines 169-170, “… the comparison cohort, matched for age, sex and county. We adjusted the models for prognostic group, CCI and educational level but not for the matching factors”.

(c). P. 8, lines 178-179, “These analyses were adjusted for prognostic group, CCI, educational level and age”.

Based on the statistics in Table 1, I assume the matched variables were sex, age, and education.

2. Furthermore, concerning some 95% CI statistics:

(a). P. 12, lines 240-241, “… however with lower unadjusted HRs than for overall mortality varying from 1.04 (95% CI 1.00-1.07)”.

(b). P. 12, line 232, “… the absolute difference at ten years was 1.4% (95% CI 0.7%-2.1%)”.

If the confidence interval crosses 1, then it implies that the HR is not statistically significant (i.e., there is no difference between arms of the study).

Kuo-Cherh Huang

Reviewers' comments:

Reviewer's Responses to Questions

**Comments to the Author**

1. Is the manuscript technically sound, and do the data support the conclusions?

Reviewer #1: Yes

Reviewer #2: Yes

2. Has the statistical analysis been performed appropriately and rigorously? 

Reviewer #1: Yes

Reviewer #2: Yes

3. Have the authors made all data underlying the findings in their manuscript fully available?

Reviewer #1: Yes

Reviewer #2: Yes

4. Is the manuscript presented in an intelligible fashion and written in standard English?

Reviewer #1: Yes

Reviewer #2: Yes

5. Review Comments to the Author

Reviewer #1: the authors reported their experience on Number of transurethral procedures after non-muscle-invasive bladder cancer and survival in causes other than bladder cancer.

the research is well pconducted and well presented.

Anymway, I suggest some rivisions:

- please state the sample size calculation

- when possible, perform a flow chart on included/exluded patients

- were patients consecutive?

- in my opinion, it is better to evaluate the risk classification rather than the grade alone of bladder cancer

Reviewer #2: The Authors investigated the overall and disease-specific risk of death in patients with NMIBC

28 compared to a background population sample. The manuscript is well written, with a good statistical analysis and explore an interesting topic. My compliments to the Authors.

6. PLOS authors have the option to publish the peer review history of their article (what does this mean?). If published, this will include your full peer review and any attached files.

Reviewer #1: No

Reviewer #2: **Yes: **Simone Sforza

---

## [Author Response · Author response to Decision Letter 0]

3 Sep 2022

PONE-D-22-17365

Number of transurethral procedures after non-muscle-invasive bladder cancer and survival in causes other than bladder cancer

PLOS ONE

Dear Dr Huang,

Thank you for a rapid and concise management of our manuscript. Please find our response below.

Journal Requirements:

Answer: We have now formatted the manuscript according to the templates and hope we have interpreted the instructions correctly.

Answer: The Research Ethics Board of Uppsala University, Sweden (Ref no. 2015/277) and Swedish Ethical Review Authority (Ref no. 2019-03574 and 2020-05123) waived the informed consent requirement for this study. This is now clarified in the manuscript.

Answer: they did not match because the statement in the manuscript did not specify who the respective PI:s were for each grant – the grant numbers were however correct. We have understood the instructions so that the funding statement should not be included in the manuscript and only in the on-line form. Therefore, the statement is excluded from the manuscript. We also tried to make the funding information clearer:

This work was supported by the Swedish Cancer Society (grant numbers CAN 2019/62 (Holmberg) and CAN 2020/0709 (Liedberg), Swedish Research Council (2021-00859) (Liedberg), Lund Medical Faculty (ALF) (Liedberg), and Skåne County Council’s Research and Development Foundation (Liedberg). The funding sources had no role in the study design, data analyses, interpretation of the results, or writing of the manuscript. Swedish Cancer Society: Cancerfonden.se; Swedish Research Council: vr.se; Lund Medical Faculty: lu.se; Skåne County: Skane.se

Answer: We are aware that this is a question often raised re Swedish register-based research and we are sorry that we were too brief in our statement. Please find below a more detailed explanation:

Data used in the present study was extracted from the research database BladderBaSe, which is based on the Swedish National Registry of Urinary Bladder Cancer (SNRUBC) and linkage to several national health-data registers. The data cannot be shared publicly because the individual-level data contain potentially identifying and sensitive patient information and cannot be published due to legislation and ethical review restrictions (https://etikprovningsmyndigheten.se). Use of the data from national health-data registers is further restricted by the Swedish Board of Health and Welfare (https://www.socialstyrelsen.se/en/) and Statistics Sweden (https://www.scb.se/en/) which are Government Agencies providing access to the linked healthcare registers. 

The data in in BladderBaSe is partly available in annual reports from the Swedish National Registry of Urinary Bladder Cancer (SNRUBC) and are also accessible online at https://statistik.incanet.se/urinblasecancer/. Researchers can propose and apply for data and studies in BladderBaSe or SNRUBC using a standardized form. After approved application, the project data administrators can upload study-specific files with selected variables to a server for statistical analysis through remote access.

Answer: These papers were accessed at PubMed the last week and there is no indication that any of them were retracted.

Additional Editor Comments:

Dear Prof. Holmberg,

We appreciate your submission to PLoS ONE. In addition to the reviewers’ comments, I would like to bring up a couple of points required your clarifications:

1. Firstly, I feel confused as regards the comparison cohort was matched on what variables exactly, as there were inconsistent statements in your paper --

(a). P. 5, lines 103-104, “Controls were matched for sex, year of birth and county of residence”.

(b). P. 7, lines 169-170, “… the comparison cohort, matched for age, sex and county. We adjusted the models for prognostic group, CCI and educational level but not for the matching factors”.

(c). P. 8, lines 178-179, “These analyses were adjusted for prognostic group, CCI, educational level and age”.

Based on the statistics in Table 1, I assume the matched variables were sex, age, and education.

Answer: The matching between the NMIBC-patients and their comparison cohort was for sex, year of birth and county of residence. We thank you for indicating that we did not word this consistently and have now done so in the manuscript. 

The analyses mentioned under (c) above were not done on the matched dataset; they did not include the comparison cohort which we now have underlined more specifically in the manuscript. Thus, these analyses did not include matching, but adjustment for prognostic group, CCI, educational level and age.

We have checked the manuscript so that we use “matching” referring to the matching between NMIBC patients and the comparison cohort and “adjusting” for when models include co-variates that potentially could be confounders.

2. Furthermore, concerning some 95% CI statistics:

(a). P. 12, lines 240-241, “… however with lower unadjusted HRs than for overall mortality varying from 1.04 (95% CI 1.00-1.07)”.

(b). P. 12, line 232, “… the absolute difference at ten years was 1.4% (95% CI 0.7%-2.1%)”.

If the confidence interval crosses 1, then it implies that the HR is not statistically significant (i.e., there is no difference between arms of the study).

Answer: Regarding (a): we have followed the convention that if the confidence interval does not encompass 1 – but only “touches” 1 - there is a sign that the null-hypothesis is statistically refuted. (In fact, the confidence interval here was 1.0028-1.07).

Regarding (b): this estimate refers to the absolute and not the relative difference and thus the point of interest in using the confidence interval as an indicator of statistical significance is 0% here. We have further stressed this in the manuscript by pointing out that the 1.4 represent percentage points of difference and does not indicate a relative measure.

Reviewer #1: the authors reported their experience on Number of transurethral procedures after non-muscle-invasive bladder cancer and survival in causes other than bladder cancer.

the research is well pconducted and well presented.

Anymway, I suggest some rivisions:

- please state the sample size calculation

Answer: To attain the highest statistical precision possible given the patients included in BladderBase, all available information in the register was used. This is now clarified in the manuscript with an example of the resulting statistical power: With 2000 events, a hazard ratio as 1.2 is detectable under standard assumptions (80 % power, two-sided 5 % significance level).

- when possible, perform a flow chart on included/exluded patients

Answer: There were no exclusions or losses to follow-up. Thus, we deem a flow-chart would be little informative, but are willing to add one if the editor so require. We have clarified in the manuscript that all patients meeting the inclusion criteria were followed.

- were patients consecutive?

Answer: Yes, the treating facilities enter patients consecutively into the SNRUBC, which is also checked against the National Cancer Register. This is now clarified in the manuscript.

- in my opinion, it is better to evaluate the risk classification rather than the grade alone of bladder cancer

Answer: We agree that it might have been informative to use the EAU risk scores for progression in our stratified analyses. This was not possible due to that the SNRUBC only recently included all the information needed for that classification and too few patients could be classified according to the EAU criteria. However, we used risk strata that underpins the clinical decision-making for Swedish NMIBC patients. This issue is now discussed under “Strengths and limitations”.

Reviewer #2: The Authors investigated the overall and disease-specific risk of death in patients with NMIBC

28 compared to a background population sample. The manuscript is well written, with a good statistical analysis and explore an interesting topic. My compliments to the Authors.

We hope that our revision addresses all the concerns raised.

With kind regards

Lars Holmberg

Prof emeritus Uppsala University and King’s College London.

---

## [Decision Letter · Decision Letter 1]

7 Sep 2022

Number of transurethral procedures after non-muscle-invasive bladder cancer and survival in causes other than bladder cancer

PONE-D-22-17365R1

Dear Dr. Holmberg,

We’re pleased to inform you that your manuscript has been judged scientifically suitable for publication and will be formally accepted for publication once it meets all outstanding technical requirements.

Kind regards,

Kuo-Cherh Huang

Academic Editor

PLOS ONE

Additional Editor Comments (optional):

Dear Prof. Holmberg,

Your rebuttal letter and re-submitted manuscript are much appreciated.  Both reviewers are positive regarding your responses.  Here, I have a remaining issue for your consideration, albeit I recommend the acceptance of your work.

One of my previous concerns was about the matched variables of your analysis as there were inconsistent statements in your paper -- (a). P. 5, lines 103-104, “Controls were matched for sex, year of birth and county of residence”.  (b). P. 7, lines 169-170, “… the comparison cohort, matched for age, sex and county. We adjusted the models for prognostic group, CCI and educational level but not for the matching factors”.  (c). P. 8, lines 178-179, “These analyses were adjusted for prognostic group, CCI, educational level and age”.  Due to the fact that frequency distributions were nearly identical between the two groups with respect to sex, age, and education in Table 1, so I would comment so at the time -- “Based on the statistics in Table 1, I assume the matched variables were sex, age, and education.”

In the rebuttal letter you had responded: “The matching between the NMIBC-patients and their comparison cohort was for sex, year of birth and county of residence. We thank you for indicating that we did not word this consistently and have now done so in the manuscript. The analyses mentioned under (c) above were not done on the matched dataset; they did not include the comparison cohort which we now have underlined more specifically in the manuscript. Thus, these analyses did not include matching, but adjustment for prognostic group, CCI, educational level and age.”

Firstly, I am not quite sure about what you meant -- “The analyses mentioned under (c) above were not done on the matched dataset; they did not include the comparison cohort [*emphasis added*] which we now have underlined more specifically in the manuscript.”  Secondly, in Table 1 the comparisons were carried out between the NMIBC cohort and the matched comparison cohort, if I am correct.  That’s why I assumed the matched variables were sex, age, and education, based on the statistics.  Indeed, in the revised manuscript you stated: “Table 1 shows the baseline characteristics of the patients in the NMIBC cohort and their 192,733 age-, sex- and county matched individuals in the comparison cohort.” (p. 9, lines 209-211).  It is clear that age is one of the matched variables.  Accordingly, the declaration seems to be problematic -- “Thus, these analyses did not include matching, but adjustment for prognostic group, CCI, educational level and age.”  Finally, I think “year of birth” and “age” would result in the same effect on your analysis, although you affirmed that the variable of year of birth was the matched variable, while age was not (that is, for the purpose of adjustment, rather -- “but adjustment for prognostic group, CCI, educational level and age”).  Then again, the argument is incompatible with the descriptor -- “age-, sex- and county-matched individuals in the comparison cohort”. 

Thank you.

Kuo-Cherh Huang

Reviewers' comments:

Reviewer's Responses to Questions

**Comments to the Author**

1. If the authors have adequately addressed your comments raised in a previous round of review and you feel that this manuscript is now acceptable for publication, you may indicate that here to bypass the “Comments to the Author” section, enter your conflict of interest statement in the “Confidential to Editor” section, and submit your "Accept" recommendation.

Reviewer #1: All comments have been addressed

2. Is the manuscript technically sound, and do the data support the conclusions?

Reviewer #1: Yes

3. Has the statistical analysis been performed appropriately and rigorously? 

Reviewer #1: Yes

4. Have the authors made all data underlying the findings in their manuscript fully available?

Reviewer #1: Yes

5. Is the manuscript presented in an intelligible fashion and written in standard English?

Reviewer #1: Yes

6. Review Comments to the Author

Reviewer #1: the authors improved the quality of the paper "Number of transurethral procedures after non-muscle-invasive bladder cancer and survival in causes other than bladder cancer" by following all the suggested revisions.

7. PLOS authors have the option to publish the peer review history of their article (what does this mean?). If published, this will include your full peer review and any attached files.

Reviewer #1: **Yes: **Luca Di Gianfrancesco

---

## [Editor Report · Acceptance letter]

15 Sep 2022

PONE-D-22-17365R1 

Number of transurethral procedures after non-muscle-invasive bladder cancer and survival in causes other than bladder cancer 

Dear Dr. Holmberg:

I'm pleased to inform you that your manuscript has been deemed suitable for publication in PLOS ONE. Congratulations! Your manuscript is now with our production department. 

Kind regards, 

on behalf of

Dr. Kuo-Cherh Huang 

Academic Editor

PLOS ONE